# Stability and change in fertility intentions in response to the COVID-19 pandemic in Kenya

**Linnea A. Zimmerman**[1]*, **Celia Karp**[1], **Mary Thiongo**[2], **Peter Gichangi**[2,3,4], **Georges Guiella**[5], **Alison Gemmill**[1], **Caroline Moreau**[1,6], **Suzanne O. Bell**[1]

**1** Department of Population Family and Reproductive Health, Johns Hopkins Bloomberg School of Public Health, Baltimore, Maryland, United States of America, **2** International Centre for Reproductive Health Kenya (ICRH-K), Mombasa, Kenya, **3** Technical University of Mombasa, Mombasa, Kenya, **4** Faculty of Medicine and Health Sciences, Department of Public Health and Primary Care, Ghent University, Ghent, Belgium, **5** Institut Supérieur des Sciences de la Population (ISSP/University of Ouagadougou), Ouagadougou, Burkina Faso, **6** Soins et Santé Primaire, CESP Centre for Research in Epidemiology and Population Health U1018, Inserm, Villejuif, France

* linnea.zimmerman@jhu.edu

**Data Availability Statement:** All data used in these analyses are publicly available and available via download upon creation of a user account at https://www.pmadata.org/data/request-access-

## Abstract

Fertility intentions are expected to decline due to the COVID-19 pandemic but limited empirical research on this topic has been conducted in sub-Saharan Africa. Longitudinal data from Kenya, with baseline (November 2019) and follow-up (June 2020) data, were used to 1) assess the extent to which individual-level fertility intentions changed, and 2) examine how security, specifically economic and health security, affected fertility intentions. The final sample included 3,095 women. The primary outcomes were change in quantum and timing. Exploratory analyses described overall changes within the sample and logistic regression models assessed sociodemographic and COVID-19 related correlates of change, specifically income loss at the household level, food insecurity, and ability to socially distance. Approximately 85% of women reported consistent fertility intentions related to both the number and timing of childbearing. No COVID-19-related factors were related to changing quantum intentions. Women who reported chronic food insecurity had 4.78 times the odds of accelerating their desired timing to next birth compared to those who reported no food insecurity (95% CI: 1.53–14.93), with a significant interaction by wealth. The COVID-19 pandemic did not lead to widespread changes in fertility intentions in Kenya, though the most vulnerable women may have accelerated their childbearing intentions.

## Introduction

The COVID-19 pandemic has impacted social and economic processes on a scale not seen in a century. Early in the pandemic, the public health community raised alarms of potentially severe consequences for reproductive health [1–5]. It was anticipated that more women would desire to avoid pregnancy due to the social and economic consequences of the pandemic, increasing demand for family planning services at a time when access to services could be negatively impacted by supply shortages, overwhelmed health systems, and restrictions on mobility [2–4]. This combination of increased demand for, reduced supply of, and reduced access to

datasets. PMA staff grant access upon creation of the account.

**Funding:** This work was supported, in whole, by the Bill and Melinda Gates Foundation through a grant (INV 009466). Under the grant conditions of the Foundation, a Creative Commons Attribution 4.0 Generic License has already been assigned to the Author Accepted Manuscript version that might arise from this submission. The funders had no role in study design, data collection and analysis, decision to publish, or preparation of the manuscript.

**Competing interests:** The authors have declared that no competing interests exist.

contraceptive services led to estimates of increases in unmet need for a modern method of contraception by between 47 million [5] and 49 million [4] women and from 7 million [5] to 15 million [4] unintended pregnancies in low- and middle-income countries (LMIC).

Current estimates now suggest, however, that increases in unmet need and unintended pregnancies were significantly lower than originally anticipated [6]. Evidence from the United States and Western Europe suggests that individuals are shifting towards delaying and limiting childbearing as a result of the pandemic, though this varies by country and across sociodemographic characteristics [7,8]. There remains, however, little empirical evidence to indicate these patterns are similar in sub-Saharan Africa, where some suggest there may actually be *increases* in desired fertility [9]. Previous research in sub-Saharan Africa has found that uncertainty, such as that introduced by the COVID-19 pandemic, whether related to economics or health, may result in increased fertility desires, as child-bearing can be viewed as a mechanism to improve stability among some populations [10,11].

Large-scale economic uncertainty has been associated with declines in both fertility intentions and fertility in the United States and Europe [12–14], though reductions are not uniform, with significant variation based on age, parity, education, and socioeconomic disadvantage [14–16] The role of economic uncertainty on fertility patterns in sub-Saharan Africa is less clear. Agadjanian found that fertility intentions in Mozambique were predicated on the stability, or instability, of individual's economic situation and shifted in response to changing economic realities [17] while Trinitapoli and Yeatman found that flexibility in fertility intentions was common among women in Malawi and served to accommodate uncertainty [10]. While these findings align with the hypothesis that uncertainty would lead to childbearing delays or limiting, other work suggests the opposite. In societies with few social safety nets, uncertainty can result in increased childbearing as a means to attain greater security [18]. Other work from Malawi also supports this; women who experienced a recent food shortage were more likely to express the desire to have a child sooner [19]. Thus, while it has been assumed that fertility intentions in sub-Saharan Africa will decline as a result of uncertainty introduced by the COVID-19 pandemic, the evidence does not necessarily support this.

Separate from economic repercussions, epidemics may introduce additional uncertainty around health and wellbeing that affects fertility, but evidence from previous epidemics is limited in its applicability to the COVID-19 pandemic. Previous work assessing the impact of the 1918–1919 influenza pandemic, the disease most similar in scale and pathology to the COVID-19 pandemic, is mixed; in the aftermath of the pandemic, fertility increased in some countries [20], declined in others [21–23], and was minimally impacted in others [24]. That the 1918 flu pandemic was contemporaneous with World War I adds additional complexity to estimating its impact on fertility. Uncertainty related to more recent epidemics, such as the HIV and Zika epidemics, has demonstrated that uncertainty related to infection and health is associated with changes in fertility patterns and preferences [10,11,25–29], but these epidemics differ significantly from COVID-19 due to the higher risk of long-term morbidity and mortality. Lessons from these more recent epidemics may thus not be applicable to the COVID-19 pandemic, which has also had a simultaneous and widescale economic impact.

## Study objective

The objective of this study is to explore how the COVID-19 pandemic, and the associated economic and health uncertainties it introduced, affected women's fertility intentions in Kenya. We focus on two demographic aspects of fertility intentions; what demographers call the quantum of fertility (i.e. the number of births women will have by the time they finish childbearing) and the tempo of fertility (i.e. the timing of these births) [30,31]. Specifically, using data from a

longitudinal study among women aged 15–49 in Kenya, this study aims to 1) describe the extent to which individual-level intentions for future children (quantum) and the time frame of intentions to have a child (tempo) change, and for whom, in the context of the onset of the COVID-19 pandemic, and 2) examine the effect that changes in economic and health security associated with the onset of the COVID-19 pandemic had on quantum and tempo of fertility intentions.

## Conceptual framework

Our analysis is guided by an adaptation to the conceptual framework created by Fahlén and Oláh to assess the role of economic uncertainty on fertility intentions in Europe [32], and Aassve and colleagues' framework anticipating the effects of COVID-19 on fertility [9] (Fig 1).

We hypothesize that the COVID-19 pandemic affects fertility intentions via multiple pathways, but ultimately operates through both the structural/societal level and the individual/household level to influence economic and health security, a critical determinant of fertility preferences. [10–12] At the structural/societal level, *societal and community factors*—including government structure and social norms such as individualism versus collectivism—influence the *social response to the pandemic*. The social response includes such actions as closing schools, imposing quarantines, and issuing stay-at-home orders. *Individual, partner, and household factors* include factors such as age, employment, and wealth, in addition to changes at the household level that may occur as a direct result of the pandemic, such as loss of employment or illness and/or the death of a family member. These factors influence an *individual's response to the pandemic* through influencing their ability and willingness to comply with health and safety recommendations, such as quarantining and social distancing. Both the social and individual responses to the pandemic influence a woman's *economic and health security*. Though other aspects of security exist, we hypothesize that in the COVID-19 pandemic, economic security and health security would be the major factors contributing to changes in fertility intentions. In addition to influencing pandemic-related behaviors, such as social distancing or masking, security will also inform *fertility intentions*, which we assess through changes in both the quantum (whether the respondent would like to have additional children) and tempo (when the respondent would like to have her next child). While not assessed in this study, fertility intentions directly affect realized fertility. We note that social factors, such as norms, and individual factors, like age and autonomy, continue to exert independent influence on fertility intentions, regardless of COVID-19. Our interest, however, is in understanding how COVID-19 affects these pathways of influence.

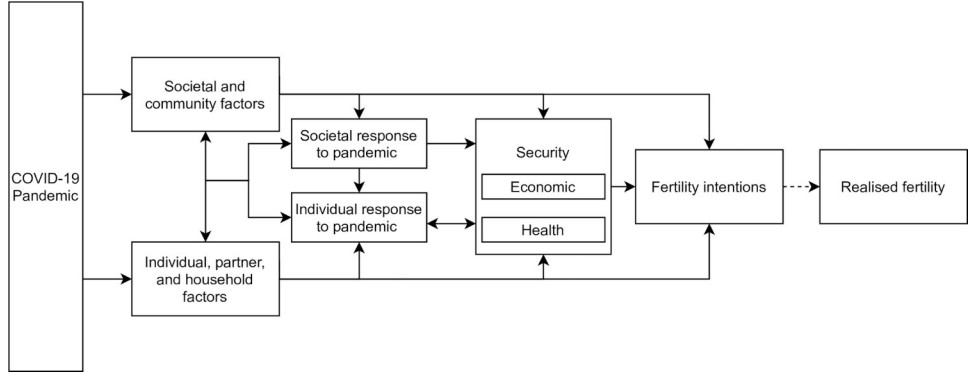

**Fig 1. Conceptual framework.**

## Methods

### Study setting

Kenya is a middle-income country in East Africa. Relative to the region, Kenya has low fertility with a total fertility rate of 3.5 [33] and high modern contraceptive prevalence at 56.4% [34]. Significant disparities exist within the country, particularly by residence and wealth, with the rural poor having significantly higher food insecurity and poor health indicators prior to COVID-19 [35]. The first confirmed case of COVID-19 in Kenya was reported on March 12, 2020 [36]. Several measures were taken within weeks of the first confirmed COVID-19 case to slow the spread of the pandemic, including social distancing and restrictions on gatherings, imposition of a night curfew, limitations on public transportation, and closure of schools [37]. At the time of follow-up data collection, detailed further below, nearly 3,000 cases of COVID-19 had been detected in Kenya, and face coverings were required in all public spaces [37]. Restrictions were strictest between mid-March and mid-June 2020 and lessened over time, however, with a resurgence in transmission beginning in mid-January, 2021, some restrictions have been re-imposed [37,38]. As of April 27, 2021, there were 156,981 cases and 2,643 deaths [39].

### Data

This study draws on longitudinal data from Performance Monitoring for Action (PMA), a panel study designed to examine key reproductive health indicators in sub-Saharan Africa and South-East Asia. PMA used a multiple stage cluster sampling approach, starting with first randomly selecting 11 counties using probability proportional to size, then randomly selecting 308 enumeration areas (EAs) using probability proportional to size, stratified by county and urban/rural residence. Within each EA, enumerators compiled a list of all households and then randomly selected 35 households per EA. A household questionnaire was administered in each household, including a roster that identified all women age 15–49 who slept in the household the night before and/or are regular members of the household. The baseline PMA panel survey was implemented in November 2019 (herein referred to as "baseline"), during which women responded to an in-person questionnaire and shared information about their sociodemographic characteristics, reproductive histories, fertility intentions, and contraceptive behaviors. Women provided written informed consent for initial interviews and provided phone numbers to participate in follow-up surveys. Parental consent and adolescent assent were obtained for women younger than age 18.

In response to COVID-19, PMA conducted a phone-based follow-up survey between May and July 2020 (herein referred to as "follow-up") during the first wave of COVID-19, when restrictions on movements and gathering were strictest. Women were eligible to participate in the follow-up survey if they owned a phone (67.7% of all women) and were successfully reached for the follow-up interview (94.9% of phone owners). Fewer than 1% of eligible women reached for follow-up refused to participate in the survey, resulting in a final sample 5,972 women (63.0% of the original sample) who completed the baseline and follow-up surveys. Due to restrictions on face-to-face interactions, all interviews were conducted over the phone and thus, women provided oral consent after interviewers read the informed consent procedures. Parental consent for adolescents was waived for the phone follow-up and adolescents provided consent. Oral consent was recorded via a checkbox in the survey by the interviewer.

**Ethics approval statement.** Ethical approval for this research was granted by the Kenyatta National Hospital-University of Nairobi Ethics Research Committee (No. P241/04/2020), and the Johns Hopkins Bloomberg School of Public Health (IRB No. 12407).

**Patient consent statement.**    All participants gave consent to participate and consent procedures approved by the Kenyatta National Hospital-University of Nairobi Ethics Research Committee (No. P241/04/2020), and the Johns Hopkins Bloomberg School of Public Health (IRB No. 12407).

## Sample

For this analysis, we restricted the sample to women who completed the baseline and follow-up surveys, were married or in-union at baseline, and reported not being pregnant or infertile at baseline or follow-up (n = 3,297). We computed post-stratification weights based on woman's likelihood of owning a phone and responding to the phone survey as a function of her age, education, wealth, and residence to account for phone ownership, non-response, and differential COVID-19 survey loss to follow-up, while adjusting for complex survey design. We excluded 202 women who stated that they did not know their fertility intentions or said that they did not have a response to the question at either baseline or follow-up as it was unclear how to define their consistency in reporting, and one woman who was missing covariate information, resulting in a final analytic sample of 3,095 women (Fig 2). There were no statistically significant differences in sociodemographic characteristics between the weighted baseline and follow-up surveys following the application of survey weights (S1 Table).

## Measures

**Outcome variables.**    Our primary dependent variable was *change in fertility intentions* during the COVID-19 period, assessed by comparing women's reported fertility intentions at baseline and follow-up. We first explored a binary measure of change related to the overall desire to have any children among nulliparous women or more children among parous women (for simplicity, we will refer to this as "any/more"), regardless of timing—we refer to this as a "quantum change". We stratified the analysis based on baseline fertility intention, creating two groups: women who reported wanting any/more children at baseline and women who reported not wanting any/more children at baseline. Among women who reported wanting any/more children, we defined women as *stable* in their fertility intentions if they consistently reported wanting any/more children at follow-up and as *antinatalist* if they shifted to not wanting any/more children at follow-up. Among women who reported not wanting any/more children at baseline, women were defined as *stable* if they reported not wanting any/more children at follow-up and as *pronatalist* if they shifted to wanting any/more children at follow-up.

We then investigated how timing of future childbearing intentions was affected by COVID-19 among women who reported wanting any/more children at baseline and follow-up–we refer to this as a "tempo change". We focused on changes in one-year fertility intentions as we hypothesize that women would be more likely to adapt their immediate childbearing intentions in response to the social and economic circumstances of the pandemic, relative to longer term fertility intentions, which may remain relatively unaffected. Similar to the quantum analysis, we stratified women based on their report at baseline, again creating two groups; women who reported wanting a child in one year or less at baseline and those who wanted to wait at least one year before the birth of their next child. Among women who stated they would like to have a child in one year or less at baseline, we classified women as *decelerators* if they shifted to wanting children later (>1 year) or *consistent* if they reported wanting children within the same time frame. Among women who indicated they wanted to wait more than one year to have a child at baseline, we classified women as *accelerators* if reported wanting a child in one year or less at follow-up. or *consistent* if they reported wanting children within the same time frame (>1 year).

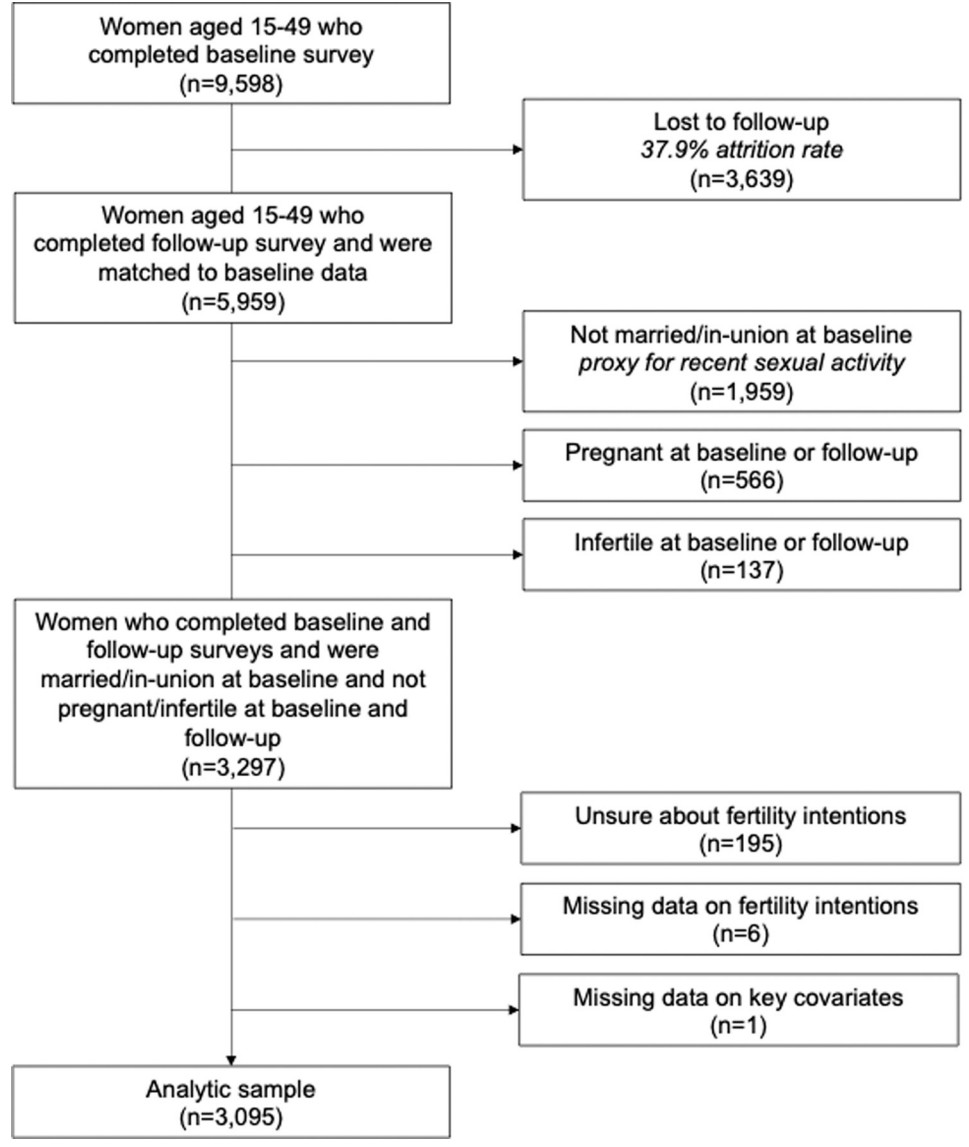

**Fig 2. Flowchart of sample selection.**

**Covariates.** Our primary independent variables assessed women's economic and health security. We initially intended to examine *perceived security* by exploring concerns about future income loss (economic security) and about being infected with SARS-COV-2 (health security) related to changes in women's fertility intentions. However, women reported nearly universal agreement to these two questions; 95.8% of women indicated being concerned/very concerned about future income loss and 92.6% were concerned/very concerned of becoming infected with SARS-COV-2, limiting the utility of these two measures. Thus, we used three alternative measures to assess security—household income loss (none, partial, complete), ability to social distance (yes/no), and food insecurity (none, chronic, increased). Ability to social distance was defined based on self-report to the question, "Are you able to avoid contact with people outside your household?"(Yes/No). Food insecurity was defined relative to pre-COVID-19 using two survey items. First, we asked if the respondent or a household member

went a whole day and night without eating since the pandemic began (yes/no), and, if so, whether this experience was more common than before COVID-19 (yes/no). We created a three-level categorical variable indicating no food insecurity, chronic food insecurity (food insecurity before and after COVID-19 that did not worsen), and increased food insecurity.

To assess the association of *individual*, *partner*, *and household characteristics*, we examined a range of sociodemographic indicators including residence (urban/rural), age (15–24, 25–34, and 35–49 years), parity (0–1, 2–3, and 4 or more children), and household wealth tertiles (low, middle, high) measured prior to the pandemic. Household wealth, which is created and provided in PMA public use datasets and was not created independently for this analysis, was measured through the creation of a continuous score from an index of household assets, using a similar methodology as employed by the Demographic and Health Survey [40]. Tertiles were created based on the household score. Nulliparous and primiparous women were combined due to sample size considerations as very few nulliparous women reported wanting no children. We treated education as a binary variable, indicating less than secondary versus secondary or higher education. All variables were selected based on the conceptual framework and were included in models independent of statistical significance.

## Analysis

**Quantum.** We first examined the quantum change in fertility intentions due to COVID-19. Descriptive analyses summarized sample characteristics and examined distributions of women's fertility intentions before and during COVID-19.

We then ran bivariate and multivariable binomial logistic regression models to explore the associations of sociodemographic characteristics and measures of security with fertility intentions at follow-up. Analysis was stratified by women's fertility intentions at baseline as we examined the odds of wanting any/more children at follow-up (*pronatal*) among women who reported wanting no/no more children at baseline, and separately assessed the odds of wanting no/no more children at follow-up (*antinatal*) women who reported wanting any/more children at baseline.

**Tempo.** Next, we examined the tempo change in fertility intentions due to COVID-19 among women who stated that they wanted to have children at baseline and follow-up. Descriptive analyses examined the proportions of women who *accelerated* or *decelerated* their fertility intentions during COVID-19 compared to baseline. We then ran bivariate and multivariable binomial logistic regression models to explore the associations of sociodemographic characteristics and measures of security with on-year fertility intentions at follow-up. Again, analysis was stratified by women's fertility intentions at baseline, as we examined the odds of wanting a child *within* one year (accelerating) among women who indicated wanting any/more children *in more than one year* at baseline, and the odds of wanting a child in *more* than one year (decelerating) among women who reported wanting any/more children *within one year* at baseline.

For both quantum and tempo, we tested for interactions between each combination of wealth, household income loss, and food insecurity, separately, to understand if wealth modified the effect of economic security on changes in fertility intentions. Only the interaction of wealth and food insecurity was significant and included in the final model.

For simplicity, Table 1 below summarizes the definitions of each group.

Multicollinearity was assessed using variance inflation factors (VIFs), which averaged 1.22 for all covariates. We excluded age, as it was moderately correlated with parity and residence, and due to the extremely small sample sizes of women <24 who expressed antinatal desires or who desired to accelerate timing. We assessed each model's goodness-of-fit via Hosmer-Lemeshow tests; none violated test assumptions (p>0.05), indicating good model fit. We defined statistical significance at p<0.05 and marginal significance at p<0.10. All analyses accounted

**Table 1. Description and operationalization of fertility intention dimensions.**

| Fertility intention dimension | Population | Status at baseline | Possible values at follow-up |
|---|---|---|---|
| Quantum | All women | Want any/more | 0: want any/more (*stable*)<br>1: want no more/none (*antinatal*) |
| | | Want no more/none | 0: want no more/none (*stable*)<br>1: want any/more (*pronatal*) |
| Tempo | Women who wanted any/more children at baseline | Want in ≤ 1 year | 0: want in ≤ 1 year (*stable*)<br>1: want in >1 year (*decelerators*) |
| | | Want in >1 year | 0: want in >1 year (*stable*)<br>1: want in ≤ 1 year (*accelerators*) |

for the complex survey design, using post-stratification weights and accounting for multistage selection and clustering. Analyses were conducted using Stata 16.1 [41].

## Results

### Sample characteristics

Characteristics of the sample are presented in Table 2. Most women lived in rural areas and had at least two children. Approximately 41% of women in Kenya completed secondary school or higher. Fewer than 10% of women Kenya reported that their households had sustained no economic losses, with half of women reporting partial loss of income. About one-quarter of women in Kenya reported heightened food insecurity during COVID-19, 69% reported being able to socially distance, and 79% reported being very concerned about becoming infected.

### Descriptive changes

**Quantum.** At baseline, 50.6% of women wanted any/more children in the future, while at follow-up, 53.6% of women indicated that they wanted any/more children in the future (Table 3). The majority of women maintained stable in their stated quantum fertility intentions between baseline and follow-up. Forty-four percent of women reported that they wanted any/more children and 40.0% reported that they did not want any/more children at both time points. Altogether, 6.5% shifted from wanting more/any children to wanting no/no more children (*antinatal*) and 9.5% switched from wanting no/no more children to wanting any/more children at follow-up (*pronatal*).

**Tempo.** Among women who consistently reported wanting any/more children at baseline and follow-up (44.1% of the total sample), 16.8% wanted a child within one year at baseline compared to 17.0% of these women at follow-up (Table 3). We found that timing of next birth remained largely stable; 75.0% of women reported wanting their next child in more than one year at both time points while 8.8% consistently reported wanting their next child in less than one year. Less than ten percent of women who wanted any/more children in both surveys *accelerated* (8.2%) or *decelerated* (8.0%) the desired timing of their next birth.

### Bivariable results

Bivariate results are shown in S1 Table. Compared to women with stable quantum intentions, only parity was associated with adopting *pronatal* intentions while education, parity, and household income loss were significantly associated with adopting *antinatal* intentions. Food

**Table 2. Characteristics of women who were married/in-union and completed the baseline and COVID-19 follow-up surveys in Kenya, PMA 2020 (N = 3,095).**

| | | | N | % |
|---|---|---|---|---|
| *Sociodemographic* | | | | |
| | **Residence** | | | |
| | | Urban | 864 | *27.9* |
| | | Rural | 2,231 | *72.1* |
| | **Age** | | | |
| | | 15–24 | 640 | *20.7* |
| | | 25–34 | 1,255 | *40.5* |
| | | 35–49 | 1,200 | *38.8* |
| | **Parity** | | | |
| | | Nulliparous | 78 | *2.5* |
| | | 1–2 | 1,137 | *36.7* |
| | | 3–4 | 1,094 | *35.3* |
| | | 5 or more | 785 | *25.4* |
| | **Education** | | | |
| | | Never | 174 | *5.6* |
| | | Primary | 1,649 | *53.3* |
| | | Secondary or higher | 1,2712 | *41.1* |
| | **Wealth** | | | |
| | | Low | 1,172 | *37.9* |
| | | Middle | 1,044 | *33.7* |
| | | High | 879 | *28.4* |
| *COVID-19-related factors* | | | | |
| | **Economic loss due to COVID-19** | | | |
| | | None | 210 | *6.8* |
| | | Partial | 1,573 | *50.8* |
| | | Complete | 1,311 | *42.4* |
| | **Concern about future income loss due to Covid-19** | | | |
| | | No | 133 | *4.3* |
| | | Yes | 2,962 | *95.7* |
| | **Food insecurity since COVID-19** | | | |
| | | None | 2,104 | *68.0* |
| | | Chronic stable | 277 | *9.0* |
| | | Increased | 713 | *23.0* |
| | **Able to socially distance** | | | |
| | | No | 954 | *30.8* |
| | | Yes | 2,141 | *69.2* |
| | **Concerned about becoming infected** | | | |
| | | Very concerned | 2,461 | *79.5* |
| | | Concerned | 408 | *13.2* |
| | | A little concerned | 84 | *2.7* |
| | | Not concerned | 141 | *4.5* |

Notes: Weighted column totals (N) and proportions (%).

insecurity was significantly associated with *accelerating* fertility intentions, while education was marginally associated. There were no observed associations between sociodemographic correlates or COVID-19 related factors with *decelerating* fertility intentions.

**Table 3. Changes to fertility intentions between baseline and follow-up, weighted proportions (%).**

| | Fertility intentions at follow-up | | | |
|---|---|---|---|---|
| **Baseline fertility intention** | **Wants more/any % (N)** | **Does not want % (N)** | **Within 1 year^ % (N)** | **More than 1 year^ % (N)** |
| Wants more/any | 44.1 (897) | 6.5 (131) | - | - |
| Does not want more | 9.5 (193) | 40.0 (811) | - | - |
| Within 1 year^ | - | - | 8.8 (79) | 8.0 (72) |
| More than 1 year^ | - | - | 8.2 (74) | 75.0 (673) |

Notes: Overall weighted proportions and totals presented by category.

^Among women who wanted children before and during COVID-19.

## Multivariable results

**Quantum.** Results of the multivariable models predicting changes in the quantum of childbearing intentions are presented in Table 4. Women with secondary or higher education and women with more than two children had significantly decreased odds of adopting *pronatal* intentions than women with less than secondary education (aOR: 0.65, 95% CI: 0.43–0.96) or women with 0–2 children (3–4 children aOR: 0.33, 95% CI: 0.22–0.50; 5+ children aOR 0.10, 95% CI: 0.06–0.16). Similarly, women with 3–4 or 5+ children had significantly higher odds of adopting *antinatal* intentions (aOR: 2.82, 95% CI: 1.84–4.28 and aOR: 6.50, 95% CI: 3.42–12.35, respectively). Rural women had 40% lower odds of adopting *antinatal* intentions than urban women, which was marginally significant (aOR: 0.60, 95% CI: 0.34–1.04). No COVID-19 related factors were significantly related with adopting *anti-* or *pronatal* fertility intentions, although the ability to socially distance was marginally positively related to *pronatal* intentions (aOR: 1.34, 95% CI: 0.97–1.85).

**Tempo.** Results of the multivariable models assessing changes in the tempo of childbearing intentions are presented in Table 5. Food insecurity was significantly related to *accelerating* fertility intentions, with women who reported chronic food insecurity having 4.78 times the odds of accelerating their desired timing to next birth compared to those who reported no food insecurity (95% CI: 1.53–14.93). COVID-19-related increased food insecurity was also marginally associated with reduced odds of accelerating fertility intentions (aOR: 0.49, 95% CI: 0.13–1.90). There were no significant associations with *delaying* fertility intentions, though chronic, stable food insecurity was marginally related to a reduction (aOR: 0.25, 95% CI: 0.03–1.78) and complete household income loss was linked an increase (aOR: 1.64, 95% CI: 0.91–2.95) in the odds of *delaying* fertility intentions. The ability to socially distance was not significantly associated with *acceleration* or *delaying fertility intentions* after accounting for other variables. Results indicated a significant interaction between wealth and changes in food insecurity during COVID-19. Specifically, the impact of food insecurity on changes in the tempo of fertility intentions varied by wealth; women who experienced chronic food insecurity were significantly more likely to accelerate childbearing relative to women who did not experience food insecurity, but this effect was significantly attenuated among women in wealthier tertiles (aOR: 0.14, 95% CI: 0.02–0.97).

## Discussion

We find that overall, there was little change in the quantum or tempo of women's fertility intentions in the early months of the COVID-19 pandemic in Kenya, despite widespread economic loss and increased food insecurity during this period. The vast majority of women

**Table 4. Adopting pronatal or antinatal childbearing intentions between baseline and follow-up (reference: Stable).**

| | | | Pronatal^ (n = 1,624) | | | Antinatal‡ (n = 1,471) | | |
|---|---|---|---|---|---|---|---|---|
| | | | aOR | 95% CI | | aOR | 95% CI | |
| *Sociodemographic* | | | | | | | | |
| | **Residence** | | | | | | | |
| | | Urban | ref | | | ref | | |
| | | Rural | 1.17 | 0.81 | 1.69 | 0.60 | 0.34 | 1.04 |
| | **Education** | | | | | | | |
| | | Primary or lower | ref | | | ref | | |
| | | Secondary or higher | **0.65**\* | **0.43** | **0.96** | 0.91 | 0.61 | 1.34 |
| | **Parity** | | | | | | | |
| | | 0–2 | ref | | | ref | | |
| | | 3–4 | **0.33**\*\* | **0.22** | **0.50** | **2.82**\*\* | **1.85** | **4.28** |
| | | 5+ | **0.10**\*\* | **0.06** | **0.16** | **6.50**\*\* | **3.42** | **12.35** |
| | **Wealth** | | | | | | | |
| | | Lowest | ref | | | ref | | |
| | | Middle | 0.88 | 0.58 | 1.34 | 1.19 | 0.68 | 2.08 |
| | | Highest | 0.82 | 0.49 | 1.37 | 1.04 | 0.53 | 2.06 |
| *COVID-19-related Factors* | | | | | | | | |
| | **Household income loss** | | | | | | | |
| | | None/Partial | ref | | | ref | | |
| | | Complete | 1.03 | 0.74 | 1.44 | 1.36 | 0.93 | 2.00 |
| | **Food insecurity since COVID** | | | | | | | |
| | | None | ref | | | ref | | |
| | | Chronic stable | 1.44 | 0.65 | 3.19 | 0.81 | 0.32 | 2.00 |
| | | Increased | 0.81 | 0.43 | 1.53 | 1.21 | 0.53 | 2.75 |
| | **Able to socially distance** | | | | | | | |
| | | No | ref | | | ref | | |
| | | Yes | 1.34 | 0.97 | 1.85 | 1.38 | 0.87 | 2.17 |
| | **Interaction: Wealth x Food insecurity** | | | | | | | |
| | | Lowest wealth x None | ref | | | ref | | |
| | | Middle wealth x Chronic stable | 1.12 | 0.39 | 3.24 | 0.81 | 0.26 | 2.59 |
| | | Highest tertile x Chronic stable | 1.20 | 0.24 | 5.93 | 0.97 | 0.25 | 3.72 |
| | | Middle tertile x Increased | 1.18 | 0.55 | 2.50 | 1.16 | 0.39 | 3.47 |
| | | Highest tertile x Increased | 1.47 | 0.57 | 3.76 | 0.61 | 0.20 | 1.85 |

Notes

^Odds of shifting to pronatal fertility intentions among women who reported wanting no more/no children at baseline.

‡Odds of shifting to antinatal fertility intentions among women who reported wanting any/more children at baseline. P-values denoted \*<0.05, \*\*<0.01.

(85%) reported consistent fertility intentions both in terms of wanting any/more children and timing of their desired next birth. We also found that COVID-19-related factors affecting economic and health security had almost no effect on the quantum of desired childbearing, though experiencing chronic food insecurity (before and during the pandemic) was associated with accelerated childbearing intentions, particularly among the poorest women. Taken together, these results provide evidence that women's fertility intentions were not as impacted by disruptions to their economic stability as research from high-income settings [7,8] or modeled estimates [1,4,5] has suggested, at least in the early months of the pandemic.

**Table 5. Acceleration and deceleration of fertility intentions among women wanting children at baseline and follow-up (reference: Stable).**

| | | | Accelerate^ (n = 1,039) | | | Delay‡ (n = 364) | | |
|---|---|---|---|---|---|---|---|---|
| | | | **aOR** | **95% CI** | | **aOR** | **95% CI** | |
| *Sociodemographic* | | | | | | | | |
| | **Residence** | | | | | | | |
| | | Urban | ref | | | ref | | |
| | | Rural | 1.10 | 0.59 | 2.06 | 0.88 | 0.43 | 1.83 |
| | **Education** | | | | | | | |
| | | Primary or lower | ref | | | ref | | |
| | | Secondary or higher | 0.69 | 0.38 | 1.26 | 1.75 | 0.83 | 3.70 |
| | **Parity** | | | | | | | |
| | | 0–2 | ref | | | ref | | |
| | | 3–4 | 0.82 | 0.44 | 1.51 | 1.55 | 0.76 | 3.15 |
| | | 5+ | 1.53 | 0.48 | 4.88 | 0.97 | 0.24 | 3.85 |
| | **Wealth** | | | | | | | |
| | | Lower | ref | | | ref | | |
| | | Middle | 1.17 | 0.48 | 2.85 | 1.23 | 0.45 | 3.38 |
| | | Highest | 1.43 | 0.54 | 3.79 | 0.50 | 0.15 | 1.69 |
| *Covid-related Factors* | | | | | | | | |
| | **Household income loss** | | | | | | | |
| | | None/Partial | ref | | | ref | | |
| | | Complete | 0.87 | 0.50 | 1.51 | 1.64 | 0.91 | 2.95 |
| | **Food insecurity since COVID** | | | | | | | |
| | | No | ref | | | ref | | |
| | | Chronic stable | **4.78**** | **1.53** | **14.93** | 0.25 | 0.03 | 1.78 |
| | | Increased | 0.49 | 0.13 | 1.90 | 1.15 | 0.30 | 4.34 |
| | **Able to socially distance** | | | | | | | |
| | | No | ref | | | ref | | |
| | | Yes | 1.00 | 0.53 | 1.91 | 1.35 | 0.73 | 2.50 |
| | **Interaction: Wealth x Food insecurity** | | | | | | | |
| | | Lowest wealth x None | ref | | | ref | | |
| | | Middle wealth x Chronic stable | **0.14*** | **0.02** | **0.97** | 1.55 | 0.11 | 21.34 |
| | | Highest tertile x Chronic stable | 0.29 | 0.06 | 1.46 | 1.31 | 0.07 | 23.03 |
| | | Middle tertile x Increased | 1.51 | 0.25 | 9.26 | 0.59 | 0.09 | 3.80 |
| | | Highest tertile x Increased | 0.92 | 0.16 | 5.23 | 0.99 | 0.13 | 7.52 |

Notes

^Odds of accelerating fertility intentions among women who reported wanting children in more than one year at baseline.

‡Odds of delaying fertility intentions among women who reported wanting children within one year at baseline. P-values denoted *<0.05, **<0.01.

The early economic impacts of COVID-19 in Kenya were substantial; unemployment rose from 5% to 21% between the last quarter of 2019 and June 2020, with disproportionate job loss among women, as wages and hourly labor fell [35]. Our data reflect this trend, with almost all women reporting either complete or partial income loss at the household level within the first few months of the pandemic and nearly universal concern about future income loss. Overall food insecurity also increased across the country [35], again reflected in our data with one-third of women reporting that food insecurity either remained stable or increased during COVID-19.

Despite these substantial economic impacts, our results indicate that COVID-19-related factors had no significant impact on the quantum of fertility intentions, that is, whether or not women reported wanting any/more children. These findings are consistent with those of Agadjanian in Mozambique, which suggest that economic insecurity may not impact overall fertility desires, at least in the short-term, but instead impact timing of childbearing [17]. Rather than shifts toward delayed childbearing, however, we found that the most economically vulnerable women, i.e. women whose families experienced chronic food insecurity and who lived in the poorest households, tended to accelerate their childbearing intentions. Though not statistically significant, a similar relationship was observed when examining shifts towards delayed childbearing intentions, wherein women in the poorest households who experienced chronic food insecurity appeared least likely to report delaying their intentions. The relationships we observed between the experience of chronic food insecurity and fertility intentions is consistent with Sennott and Yeatman's observations in Malawi, suggesting childbearing may function as a means of securing financial stability from partners and/or family members and as protection from further economic hardship [19]. In contrast, the finding that women who experienced increased food insecurity during COVID-19 were marginally less likely to accelerate childbearing appears contradictory. This may be partly due to the fact that our measure of familial food insecurity lacked specificity and may include circumstances in which women and their families may temporarily run out of food in the absence of sustained economic hardship. This is supported by the fact that 24% in the richest households reported increases in food insecurity during COVID-19. Our finding of a significant interaction between the effect of food insecurity and wealth, wherein the effect of food insecurity was significantly attenuated among wealthier women, further supports this explanation.

Beyond COVID-19 related circumstances, we found that flexibility in fertility intentions was also dependent on women's parity. Women who had more children may be more likely to have already attained their ideal family size and, thus, are less likely to shift their intentions to wanting more children in response to external circumstances. In contrast, lower parity women, who are also generally younger, retain greater flexibility in their fertility intentions and may be more impacted by the circumstances induced by the pandemic.

Ultimately, fertility intentions inform women's need and demand for reproductive and maternal health services. Models predicting large increases in unintended pregnancy anticipated a rise in demand for family planning services, coupled with a decline in access; however, in our sample, with largely stable fertility intentions and modest shifts towards wanting children sooner, demand for these services is not likely to increase. This is consistent with recent population-level findings in four sub-Saharan African geographies, reporting sustained need for contraception in the early stages of the pandemic, relative to trends observed before the onset of COVID-19 [42]. Similarly, Karp and colleagues found that most women in Kenya (81.6%) did not change their contraceptive status in the early months of the pandemic, and those who did were more likely to adopt a method than to discontinue (13.1% vs. 5.3%, respectively) [43]. These findings indicate that, at least in the early stages of the pandemic, family planning services were not adversely affected on a large scale. Similarly, our findings indicate that need for maternal health services will not decrease markedly during the pandemic. Recent evidence suggests that COVID-19 carries an increased risk for maternal and newborn morbidity and mortality [44,45]. Communicating about this potential increased risk, in addition to counseling on effective COVID-19 prevention strategies, including the safety of receiving the COVID-19 vaccine during pregnancy [46,47], should be prioritized as part of maternal health care services.

## Strengths and limitations

This study has a number of strengths. It is unique in that we were able to use longitudinal data to assess the early impact of COVID-19 on fertility intentions. By linking women and comparing their fertility intentions across two time points we were able to eliminate recall bias that may otherwise result from women retrospectively reporting their intentions before and during the pandemic. Due to the short time-period between baseline and follow-up, it is likely that the majority of changes are a result of the ubiquitous social and economic disruption that occurred, rather than widespread changes to demographic events, such as marriage and child-bearing, which would generally occur over longer periods of time. The short time frame between baseline and follow-up also reduced the potential for misclassification of fertility timing, though it does not eliminate the potential for it to affect our results. More specifically, some women may have reported at baseline that they wanted a child in more than a year, for example in 15 months, but when asked four months later, reported that they wanted a child in 11 months. These women would be consistent in their report but classified as accelerators in our analysis. However, given the relatively short time period between these surveys, we believe the potential for this misclassification on a large scale is unlikely. Additionally, we were able to use data from individual and household reports of the direct impact of COVID-19 on social, health, and economic circumstances brought about by the pandemic, providing further nuance to our understanding of the mechanisms through which COVID-19 may impact fertility intentions.

However, there are limitations worth noting. First, our sample was restricted to women who could be recontacted via phone. While most women in Kenya own phones, and we used post-stratification weights to account for differences in the probability of owning a phone and being interviewed at follow-up, it is possible that we were unable to account for all factors associated with phone-owner-based exclusion. If not, results may not be generalizable to those with fewer resources, who may be the most impacted by economic uncertainty. Secondly, we excluded women who reported that they did not know if and when they would like to have another child at either interview as we could not classify these women as pronatal/antinatal or accelerators/decelerators for analysis. Though this was not a large percentage of women (approximately 6%), it is possible that these women differ systematically from women with more codified intentions. We found that these women were slightly older and of higher parity; their exclusion may thus modify the relationship we found between women's parity and changes to the quantum of their fertility intentions. Given the strength of this relationship, however, it is unlikely that such sample differences would change our conclusions. Still, this is an important area for future exploration. Hayford found that uncertainty related to mortality and economic conditions was predictive of higher reporting of non-numeric responses in Mozambique, reflecting that greater uncertainty at a social level results in greater uncertainty at the individual level [48]. While we could not explore this in great detail due to our limited sample size, future research should explore whether the consequences of the COVID-19 pandemic, specifically higher mortality and economic uncertainty, result in greater fertility ambiguity in the population. Finally, our study was conducted during the first three to four months of the pandemic using phone-based surveys. Though there were obvious economic and social impacts at the national level even in this time period, it is unclear if these patterns will generalize to later periods in the pandemic. That is, prolonged exposure to the COVID-19 pandemic and the associated global recession could result in different dynamics over time. This relationship can be further explored with ongoing PMA data collection efforts in Kenya, in addition to several other sub-Saharan African countries. We also note that using two different modes of data collection (phone versus face-to-face) may introduce differential responses based on

mode of interview; however, as all respondents had face-to-face interviews first and phone-based interviews second, it is not possible to quantify any potential bias.

## Conclusion

Insecurity introduced by the COVID-19 pandemic did not lead to widespread changes in fertility intentions in Kenya, though evidence suggests that the most vulnerable women accelerated fertility intentions in response to the pandemic. Understanding how wide-spread changes in economic security affect fertility intentions is critical, as these intentions shape demand for reproductive and maternal health services.

## Supporting information

**S1 Table. Sample characteristics at baseline and follow-up for weighting procedures.**
(DOCX)

**S2 Table. Bivariate associations between women's characteristics and changes in fertility intentions.**
(DOCX)

## Acknowledgments

We would like to thank Phil Anglewicz, Linnea Eitmann, and Beth Larson for their thoughtful review and suggestions for this manuscript. Additionally, we thank the PMA Kenya team for their tireless efforts to field the Performance Monitoring for Action (PMA) surveys, before and during the COVID-19 pandemic and the many respondents that contributed their time and responses to these surveys. Finally, we would like to recognize the more than three million people who have lost their lives to COVID-19.

## Author Contributions

**Conceptualization:** Linnea A. Zimmerman, Celia Karp, Alison Gemmill, Caroline Moreau, Suzanne O. Bell.

**Data curation:** Mary Thiongo.

**Formal analysis:** Linnea A. Zimmerman, Celia Karp.

**Investigation:** Linnea A. Zimmerman, Mary Thiongo, Peter Gichangi.

**Methodology:** Linnea A. Zimmerman, Mary Thiongo, Peter Gichangi, Georges Guiella.

**Project administration:** Mary Thiongo, Georges Guiella.

**Supervision:** Linnea A. Zimmerman, Peter Gichangi.

**Writing – original draft:** Linnea A. Zimmerman, Celia Karp, Alison Gemmill, Caroline Moreau, Suzanne O. Bell.

**Writing – review & editing:** Linnea A. Zimmerman, Celia Karp, Mary Thiongo, Peter Gichangi, Georges Guiella, Alison Gemmill, Caroline Moreau, Suzanne O. Bell.

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
