## [Decision Letter · Decision Letter 0]

8 Oct 2021

PGPH-D-21-00172

Title: A longitudinal exploration of stability and change in fertility intentions in response to the COVID-19 pandemic in Kenya

Dear Dr. Zimmerman, 

Thank you for submitting your manuscript to PLOS Global Public Health. After careful consideration, we feel that it has merit but does not fully meet PLOS Global Public Health’s publication criteria as it currently stands. Therefore, we invite you to submit a revised version of the manuscript that addresses the points raised during the review process.

We look forward to receiving your revised manuscript.

Kind regards,

Rajat Das Gupta, M.D.

Academic Editor

Journal Requirements:

1. During our internal checks, the in-house editorial staff noted that you conducted research or obtained samples in another country. Please check the relevant national regulations and laws applying to foreign researchers and state whether you obtained the required permits and approvals. Please address this in your ethics statement in both the manuscript and submission information. In addition, please ensure that you have suitably acknowledged the contributions of any local collaborators involved in this work in your authorship list and/or Acknowledgements. Authorship criteria is based on the International Committee of Medical Journal Editors (ICMJE) Uniform Requirements for Manuscripts Submitted to Biomedical Journals.

2. We note that participants provided oral consent. Please state in the Methods:

- Why written consent could not be obtained

- Whether the Institutional Review Board (IRB) approved use of oral consent

- How oral consent was documented

- Whether consent was informed.

3. You indicated that you had ethical approval for your study. In your Methods section, please ensure you have also stated whether you obtained consent from parents or guardians of the minors included in the study or whether the research ethics committee or IRB specifically waived the need for their consent.

4. Please include additional information regarding the survey or questionnaire used in the study and ensure that you have provided sufficient details that others could replicate the analyses. For instance, if you developed a questionnaire as part of this study and it is not under a copyright more restrictive than CC-BY, please include a copy, in both the original language and English, as Supporting Information.

5. Please provide separate figure files in .tif or .eps format only, and remove any figures embedded in your manuscript file.  If you are using LaTeX, you do not need to remove embedded figures.

6. Tables should not be uploaded as individual files.  Please remove these files and include the tables in your manuscript file.

7. We notice that your supplementary figures are uploaded with the file type 'Figure' and are therefore included in the PDF.   Please amend the file type to 'Supporting Information'  . Please ensure that all Supporting Information files are included correctly and that each one has a legend listed in the manuscript after the references list.

8. We have noticed that you have uploaded supporting information but you have not included a list of legends.  Please add a full list of legends for all supporting information files (including figures, table and data files) after the references list. 

9. The link you provided in the Data Availability Statement returns no result . Please amend your Data Availability Statement and indicate where the data may be found.

Additional Editor Comments (if provided):

Dear Authors,

Thank you for submitting the manuscript in Plos Global Public Health. Before proceeding further, please address the reviewers’ comments as well as the following comments:

1. Please mention in which wave of COVID-19 pandemic in Kenya the research was conducted? Due to the rapidly changing nature of the pandemic this is very important to mention the wave of the pandemic.

2. Baseline data was collected by in person questionnaire and follow-up survey was done using telephone interview. As the authors employed two types of data collection technique, does it have any implications on the nature of information bias in the study?

3. Please mention how the authors adjusted the survey weight and control for cluster effect during the analysis.

4. Please mention how variables was selected in the multivariable binomial logistic regression?

5. The study was conducted at the early part of pandemic and the findings do not reflect their intention as the pandemic progresses. This should be mentioned in the limitation.

Reviewers' comments:

Reviewer's Responses to Questions

**Comments to the Author**

1. Does this manuscript meet PLOS Global Public Health’s publication criteria? Is the manuscript technically sound, and do the data support the conclusions? The manuscript must describe methodologically and ethically rigorous research with conclusions that are appropriately drawn based on the data presented.

Reviewer #1: Yes

Reviewer #2: Partly

Reviewer #3: Yes

Reviewer #4: Partly

Reviewer #5: Yes

Reviewer #6: Partly

2. Has the statistical analysis been performed appropriately and rigorously?

Reviewer #1: Yes

Reviewer #2: Yes

Reviewer #3: Yes

Reviewer #4: No

Reviewer #5: Yes

Reviewer #6: No

3. Have the authors made all data underlying the findings in their manuscript fully available (please refer to the Data Availability Statement at the start of the manuscript PDF file)?

Reviewer #1: Yes

Reviewer #2: Yes

Reviewer #3: Yes

Reviewer #4: Yes

Reviewer #5: Yes

Reviewer #6: Yes

4. Is the manuscript presented in an intelligible fashion and written in standard English?

Reviewer #1: Yes

Reviewer #2: No

Reviewer #3: Yes

Reviewer #4: Yes

Reviewer #5: Yes

Reviewer #6: Yes

5. Review Comments to the Author

Reviewer #1: I would like to congratulate the authors for the submission—I find the article clear, to the point, and insightful as to what is happening in a country of Sub-Saharan Africa in regards to fertility intentions during the COVID-19 pandemic. This study fills a critical knowledge gap, as the available research on this issue mainly focuses on high-income countries. The findings of this study would be considered useful not only for Kenya but also for other developing countries.

The statistical methods used in the manuscript are appropriate (including the calculation of the post-stratification weight). My major comments on the analysis, which can be addressed easily, are as follows:

‒ The 1st COVID-19 case was confirmed on 12 March 2020, and the follow-up survey was conducted during May-July 2020. Can the authors elaborate on the timeline of COVID-related restrictions in Kenya (viz., cessation of movement in and out of major cities, school and/or business closures, lockdown/curfew, etc.) to clarify if the major restrictions were put into place before or after the follow-up survey.

‒ In total, 36% of the baseline respondents were not included in the follow-up. The authors consider including a supplementary table to compare their basic characteristics to the analytic sample. If these two groups are similar, then the generalizability of the findings would be strengthened.

‒ The authors tested for interactions between each combination of wealth, household income loss, and food insecurity. Tables 4 and 5 only listed Wealth x Food insecurity interaction term. Is this because the Wealth x HH income loss interaction term was not significant in the “test”?

‒ Tables 4 and 5 should include a marker for marginally significant (p<0.10) estimates since marginally significant results were highlighted in the text.

Minor comments:

‒ For the ease of the readers, Table 1 can include predictors used in the analysis as well.

‒ Citation style and formatting require review.

‒ Tempo sub-section in results (pp.14-15, lines 359-366) needs to refer to Table 3.

‒ The manuscript needs a review for correcting typos (e.g., p.16, line 386).

Reviewer #2: This manuscript goes mostly in line with PLOS Global Health since this is an original research, not published elsewhere and attempted a robust analysis with available data. However, I believe, there have been some misinterpretation of the results in the paper. The result and discussion section needs revision. But, more importantly, the timeline of the study does not seem to be sufficient to address the said objectives. Because the follow up time did not cover even any of the four Covid waves completely. The data were collected when the first wave just started. To me this is a big concern.

Reviewer #3: The manuscript is timely and relevant to a contemporary problem; Also, it is well written, logical, and easy to understand; and the study is well designed and employs acceptable methods. There are a few small details that require fixing.

1. In the manuscript, there is no definition of household wealth or ability to socially distance oneself.

2.In order for other researchers to repeat the experiment, household food insecurity must also be described.

3. Instead of Table 2, the authors might include a bivariate table that includes women's characteristics and baseline fertility intentions.

Reviewer #4: Comments for the Authors

This manuscript investigates the effects of the COVID-19 pandemic on fertility intentions. The study was conducted using data from the Performance Monitoring for Action (PMA), a longitudinal data collected to examine key reproductive indicators in sub-Saharan Africa and South-East Asia. The current prospective study used the PMA data collected in November, 2019 and follow-up in June 2020 and examined how economic and health security due to the pandemic affected the change in quantum and timing of fertility intentions of 3, 095 women in Kenya. Exploratory analysis was used to describe the overall changes in the sample whilst logistic regression models were used to assess sociodemographic and COVID-19 related changes. The study reported no COVID-19 related factors associated quantum of childbearing. However, women who reported chronic food security had 4.78 times higher odds of accelerating their desired timing to next birth compared to those who reported no food insecurity. The study concluded that the COVID-19 pandemic did not lead to widespread changes in fertility intentions in Kenya, though vulnerable women may have accelerated their childbearing intentions.

The study addresses an important issue on the impact of the CoVID-19 pandemic in relation to women’s intention to give birth to more and the timing of these births. The study utilized data from the Performance Monitoring for Action survey which was conducted both before and during the CoVID-19 outbreak in Kenya. The study has key strengths including the robust design of the data collection process and the prospective nature of the data collected. However, the study has notable weaknesses that are worth addressing. These include:

1. The study’s title is misleading. The title talks about longitudinal exploration, and this is repeated in the discussion. However, the investigators applied logistic regression in their analysis. For a longitudinal exploration one would expect to see profile analysis, Generalized Estimating Equations (GEE) or mixed models as these will help explore the longitudinal as well as any cross-sectional effects. The investigators should clarify the use of longitudinal exploration in the title and discussions.

2. The use of ordinary logistic regression limits the ability to assess the longitudinal effects of the impact of COVID-19 on the fertility intentions. It may be important to examine these using GEE. The investigators need to justify why they did not use longitudinal methods.

3. The loss to follow up of 37.9% is so high to ignore. The investigators should explore and report, just like they did with the 6% of the women who reported they did not know if and when they will have another child, how those who loss to follow up differed from those who were part of the final analysis.

4. The CoVID-19 pandemic outbreak occurred mostly in some urban areas in Kenya. It will be important to account for the areas where the study participants are coming from in relation to how severe the pandemic was. For instance, what’s the distribution of the study participants between the areas worst hit by the pandemic and those that are not? From the Table 2, it will appear that most of the study participants (72.1%) were from the rural areas. It will seem to suggest that since most of the hard hit areas were the urban centers, most participants in the study may not have really been affected by the pandemic. There is the need to account for this variability.

5. Table 2 needs some further explanation. The table presents weighted columns totals and proportions. One would expect that since both measures are weighted, they will tally. However, the proportions do not tally with the totals. Please address this and present a clearer table.

Reviewer #5: The topic of the paper is much interesting and timely given the global and regional context of the COVID 19 scenario. This paper would also have significant influence of reproductive health policies and programs.

The introduction section, in general, looks good. However, I do emphasize on tightening the section and focusing more on the points of interest rather than having a detailed literature review. The methodology, result and discussion sections are well presented and elaborated. Some specific comments are as follows:

Line 93: Please be specific what was anticipated.

Line 423: “research in high-income settings has suggested,”, need reference for this claim

Reviewer #6: The study has several strengths and significant implications. However, there are many key issues that should be addressed.

1. The title is misleading. In this study, longitudinal data were used to define the stability and change in fertility intentions variables. Indeed, instead of showing the longitudinal trend (e.g., over time such as Nov 2019 – June 2020, July–Dec 2020, Jan–June 2021, etc.), the authors explored the factors associated with stability and change in fertility intentions.

2. The Introduction is unnecessary long. The authors discussed so many things but did not explain the appropriate context in Kenya.

3. PMA used a multiple-stage cluster sampling approach. Please mention the exact number of stages and elaborate on how data collection was done in each stage.

4. Please use separate heading to define the outcome(s) and covariates.

5. The methods used are technically incorrect. The authors did not appropriately use the survey features (e.g., sampling weights, strata, and clusters).

6. Marginally associated is an awkward term. Suggest removing this term. Please interpret the results using the point estimate and 95% CI instead of focusing on p-values. The authors focused too much on statistical significance, which relies on sample size. However, some estimates seem to be clinically relevant (e.g., the odds ratio of 0.25), but the authors did not pay careful attention to explain these estimates.

7. Odds ratios seem to be overestimated, and some 95% intervals are uninterpretable. I suggest reporting the risk ratio. Besides, please report the risk difference (at least in the appendix), which is more important for target interventions.

8. Model diagnostic is missing, which is very relevant in the context of this study.

9. The first sentence of the Discussion doesn’t make any sense. If the authors refer to Table 3 for this sentence, I should say the results are confounded. Besides, widespread economic loss and increased food insecurity were not measured in this study but supported from the literature.

10. Practically speaking, there could be many time-varying covariates that influence the stability and change in fertility intentions. Even some variables considered in this study (e.g., food insecurity) could be time-varying. The authors should expand the context in the literature and limitations parts discussing this time-varying covariate issue.

11. Overall, the study aims vs. the results authors showed were inconsistent.

11a. The authors aimed to “assess the extent …” but showed the confounded results in Table 3. That means the authors only showed the unadjusted incidence of changes to fertility intentions. Exploring the incidence of changes to fertility intentions could be one aim, but it should be clear to the readers before showing the results.

11b. The authors aimed to “examine how changes …” but actually explored the factors associated with stability and change in fertility intentions. The term “how” in the aim made it ambiguous, or at least as a reader, I didn’t understand the authors’ explanation.

Some minor comments include:

• Please explain how the household wealth variable was constructed.

• Results in Table 3 are unadjusted and thus confounded. Please provide the adjusted estimates (at least age and complex design-adjusted estimates).

• Please use “multivariable model” instead of “multivariate model”.

• Please use “x times odds” or “(x-1)*100% higher odds”, not “x times higher”.

• Tables 4 and 5 could be improved. The authors could show the interaction estimates for each category (e.g., within lowest, none vs. chronic stable and none vs. increased, and within none, lowest vs. middle, etc.). The Publish package in R could be helpful.

6. PLOS authors have the option to publish the peer review history of their article (what does this mean?). If published, this will include your full peer review and any attached files.

**Do you want your identity to be public for this peer review?** For information about this choice, including consent withdrawal, please see our Privacy Policy.

Reviewer #1: No

Reviewer #2: No

Reviewer #3: **Yes: **Ahmed Hossain

Reviewer #4: No

Reviewer #5: **Yes: **Abdullah Nurus Salam Khan

Reviewer #6: No

---

## [Decision Letter · Decision Letter 1]

19 Jan 2022

Stability and change in fertility intentions in response to the COVID-19 pandemic in Kenya

PGPH-D-21-00172R1

Dear Dr. Zimmerman,

We're pleased to inform you that your manuscript has been judged scientifically suitable for publication and will be formally accepted for publication once it meets all outstanding technical requirements.

Within one week, you'll receive an e-mail detailing the required amendments. When these have been addressed, you'll receive a formal acceptance letter and your manuscript will be scheduled for publication.

An invoice for payment will follow shortly after the formal acceptance. To ensure an efficient process, please log into Editorial Manager at https://www.editorialmanager.com/pgph/ click the 'Update My Information' link at the top of the page, and double check that your user information is up-to-date. If you have any billing related questions, please contact our Author Billing department directly at authorbilling@plos.org.

Kind regards,

Rajat Das Gupta, M.D.

Academic Editor

Additional Editor Comments (optional):

Reviewers' comments:

Reviewer's Responses to Questions

**Comments to the Author**

1. If the authors have adequately addressed your comments raised in a previous round of review and you feel that this manuscript is now acceptable for publication, you may indicate that here to bypass the “Comments to the Author” section, enter your conflict of interest statement in the “Confidential to Editor” section, and submit your "Accept" recommendation.

Reviewer #1: All comments have been addressed

Reviewer #3: All comments have been addressed

Reviewer #6: All comments have been addressed

2. Does this manuscript meet PLOS Global Public Health’s publication criteria? Is the manuscript technically sound, and do the data support the conclusions? The manuscript must describe methodologically and ethically rigorous research with conclusions that are appropriately drawn based on the data presented.

Reviewer #1: Yes

Reviewer #3: Yes

Reviewer #6: Yes

3. Has the statistical analysis been performed appropriately and rigorously?

Reviewer #1: Yes

Reviewer #3: Yes

Reviewer #6: Yes

4. Have the authors made all data underlying the findings in their manuscript fully available (please refer to the Data Availability Statement at the start of the manuscript PDF file)?

Reviewer #1: Yes

Reviewer #3: Yes

Reviewer #6: Yes

5. Is the manuscript presented in an intelligible fashion and written in standard English?

Reviewer #1: Yes

Reviewer #3: Yes

Reviewer #6: Yes

6. Review Comments to the Author

Reviewer #1: I thank the authors for incorporating all the comments.

Reviewer #3: (No Response)

Reviewer #6: The paper is in good shape and can be accepted for publication at the PLOS Global Public Health. Congratulations to the authors!

7. PLOS authors have the option to publish the peer review history of their article (what does this mean?). If published, this will include your full peer review and any attached files.

**Do you want your identity to be public for this peer review?** For information about this choice, including consent withdrawal, please see our Privacy Policy.

Reviewer #1: No

Reviewer #3: **Yes: **Ahmed Hossain

Reviewer #6: No
